# Impacts of environmental uncertainty on investment stocks perception under the holiday effect

Shih-Yung Wei[1], Li-Wei Lin [2]*

**1** School of Mathematics & Statistics, Shaoguan University, Shaoguan City, Guangdong, China, **2** College of Business Administration, Fujian Jiangxia University, Fuzhou City, Fujian, China

* linlw1982@gmail.com

**Data Availability Statement:** We use Taiwan TEJ database source.(Taiwan's largest database website). https://www.tej.com.tw/.

**Funding:** The authors received no specific funding for this work.

## Abstract

This study explored how the holiday effect impacts the fluctuations in various scale indexes. Using differential and double-difference methods, the researchers of this study analyzed the impact of the lockdown in Wuhan, China on the holiday effect during the COVID-19 pandemic. The research objects used in this study include CSI All Share, CNI1000, CNI 2000, CNI Large Cap., CNI Mid-Cap., and CNI Small Cap. This study found that on behalf of the Chinese market index and the large, medium, and small-scale index, stock volatility is evident on the next day following successive holidays. Meanwhile, greater volatility is observed in small stocks' 4-day vacation (May 1, 11) than in a two-day vacation. The researchers discovered that the sealing effect causes investors to feel uncertain about the increased stock volatility. In terms of size, the net impact of the pandemic on the stock holiday effect is also greater for small stocks than for large stocks. This study's main contribution is the GARCH +DID hybrid method.

## 1. Introduction

COVID-19's sudden emergence in mainland China forced citizens to stay home while still being required to report to work and attend classes remotely. Despite the sudden changes in the citizens' daily activities due to the lockdown brought about by the COVID-19 pandemic, the stock market continues its operations, allowing investors to trade and buy stocks online.

The efficient market hypothesis asserts that if a security market is efficient, security prices should fully and immediately reflect all relevant information. However, many empirical results in recent years have found that the futures market price has some abnormal phenomena related to time, which is quite different from the efficient market hypothesis of the security market. The abnormal phenomenon represents a regular change in the financial market, and it becomes an important indicator for investors to obtain excess returns and avoid risks. In general, time-dependent anomalies are classified as follows: day effect, overnight effect, weekend effect, month effect, and January effect. The holiday effect is primarily responsible for the weekend effect and the January effect.

In the 1960s, the study of asymmetric information gave rise to signal theory. Investors face numerous investment risks due to information asymmetry. Beretta and Bozzolan (2004)

**Competing interests:** The authors have declared that no competing interests exist.

pointed out that it is critical for investors to consider the company's risk disclosure during the investment process [1]. Elshandidy, Fraser, and Hussainey (2013) mentioned that risk disclosure is related to the company's risk [2].

The COVID-19 pandemic has caused panic and environmental uncertainty among market investors. Different modes of consumption were introduced, along with inflation, monetary easing policy, and rising interest rates, among others. The introduction of these signals caused investors to risk their investments. Considering that this information will impact investors' investment decisions, the researchers of this study aim to determine investors' ability to cope with higher investment risks. The Federal Reserve Board of the United States announced in June 2022 that it expected to raise interest rates by three cents due to inflation, causing a sharp drop in the Dow Jones stock market and investors' confidence and risk appetite. As a result of inflation, investors preferred to hold their money, and they also considered the risks that other investors experienced in the stock market.

Many stock investments have observed increased risk and uncertainty due to the COVID-19 pandemic. Taiwan's Ruentex Group was affected by the US interest rate hike in 2022 and suffered losses after securing US dollar bonds. Indirectly, the stock price of the entire Ruentex Group dropped for three consecutive days. Many uncertain market factors can influence investors' decisions, and the presence of risks can affect subsequent stock prices. Narayan (2020A) mentioned that the impact of the COVID-19 pandemic resulted in more information being generated in the market [3].

The researchers of this study investigated the data of A-share listed companies in China. The small and large stocks in China were then analyzed after collecting a large amount of data. According to Benartzi et al. (2007), stock price movements have become a normal relationship [4]. Considering the changes in investors' risk appetite and investment decisions during the COVID-19 pandemic, the researchers of this study raise the question: Is it due to the stock market's volatility during the pandemic?

This type of research study is the first in mainland China, which can broaden the scope of market research in the country. The researchers of this study addressed the gaps in previous stock price research by taking into account specific factors, such as the environment and war.

The study used an innovative research method, combining GARCH and DID. Using this method, the researchers aim to strengthen the prediction correctness of the whole research.

In this study, the researchers solved the final accuracy technology for predicting stock prices, allowing investors to make more accurate decisions. We can summarize the factors that influence the share price from Table 1 Illustrative research.

## 2. Literature review

### 2.1 Social capital theory and signaling theory

Since the 1970s, social capital theory (also known as total social capital) has been studied in many disciplines, such as economics, sociology, behavioral organization theory, and political science. The impact of COVID-19 and the war in Europe is well-known, leading to uncertain impacts on global stock markets. According to Lin et al. (2022), the concept of intelligent capital belongs to intangible assets, and many investors disregard the investment signal of intangible assets [12]. Due to the COVID-19 pandemic, many investors are starting to look at medical concept stocks. However, investors considering medical-related stocks must start paying attention to their overall R&D and knowledge management. Mohammad Alghababsheh et al. (2021) explored the concept of capital in relation to relationships and cognition. When investors trade related stocks, they make investment decisions based on the concept of previous investment cognition [13]. Li et al. (2014) mentioned that social capital is derived from the

**Table 1. Illustrative research summarizing the factors that influence the share price.**

| Illustrative research | Context | Theoretical basis | Antecedents to share price | Key findings or propositions |
|---|---|---|---|---|
| Fama (1970) [5] | Information share prices soon after it is discovered | Signaling theory | Timing | The findings suggested that timing and likelihood of occurrence delivered a contribution to the company's residual value |
| Malcolm Baker et al.(2008) [6] | Proposed nominal stock prices catering theory | Catering theory | Time-series | The findings revealed that there is empirical support for the predictions in both time-series and firm-level data. |
| Dolley (1933) [7] | Impact of nominal prices | Trading costs depend on nominal prices theory | Timing | The findings suggested that investors refer to nominal (per-share) stock price. |
| Green and Hwang (2007) [8] | Linkage between stock price and low-priced stocks | Expectation theory | Time-series | The results revealed that stocks that split experience sudden increases in comovement with low-priced stocks. |
| Ohlson and Penman (1985) [9] | Splitting stocks comove more with low-price and generally smaller-cap stocks | Expectation theory | Time-series | The results show that stocks that split experience greater volatility increases. |
| Dyl and Elliott (2006) [10] | Cross-sectional stock price research | Decision theory | Time-series | The findings suggested that there is a strong cross-sectional relationship between size and share price. |
| Black (1986) [11] | Low price impact | Decision theory | Time-series | Further, investors view low-priced stocks as subject to greater noise trading. |

trading relationship and information-sharing in the market, which includes the establishment of cognition and relationship [14]. Meanwhile, Carey et al. (2011) mentioned that social capital comes from information-sharing and goals [15]. Many investors look at external information and content to evaluate the content of their investments. Moreover, many investors look at the fundamentals of individual stocks to decide which stocks to target. Backer (2010) mentioned that environmental factors send some signals to investors, so they can decide whether to invest [16].

The impact of the entire COVID-19 pandemic caused many investment risks to surface. In 2022, the world experienced inflation, causing many investments to become risky. Kim et al. (2011) mentioned that financial transparency influences investors' final investment decisions [17]. Li (2006) mentioned that the opacity of financial reports affects subsequent investors' risk appetite in making stock investment decisions [18]. Considering the concept of information asymmetry, the emergence of the COVID-19 pandemic caused environmental uncertainty and a greater increase in investment risk to surface.

## 2.2 Stock risks and contingencies

Unexpected events can occur in the stock market, and investors must consider these risks. Mossin (1966) discussed the influence of the stock system on the market [19]. Ross (1976) pointed out that risks come from the macroeconomic impact, which has an indirect effect on stock prices [20]. Roll (1988) mentioned that the external environment affects the stock's speculative change ability of stocks [21]. The impact of COVID-19 has caused a lot of investment volatility in the stock market. In 2022, the world was affected by the war in Europe, t global inflation, and the big risk of stock investment. According to Debondt and Thaler (1989), the stock price falls due to fluctuations in the external environment but eventually returns to the average stock price value over time [22]. The impact of the pandemic has led to volatility in China's stock market, and these risks influenced investors' risk appetite.

## 2.3 Environmental uncertainty factors

According to Pastor and Veronesi (2013), overall economic policy insecurity will affect the investment and development of the stock market as a whole [23]. Brogaard and Detzel (2015) mentioned that economic policy insecurity would increase the overall stock market investment risk [24]. Li et al. (2020) mentioned that China's stock fluctuates due to its overall economic policy [25]. The overall economic policy will influence the impact of the entire environment on the stock price. With the impact of the pandemic, the overall environment has become more uncertain, especially in 2022. Inflation affected the economic environment of many countries, and the whole economy is worsening. Due to this, many of the underlying stock market trends are worth studying. Kahneman and Tversky (1979) mentioned that external environmental factors influence investors' decisions on which stock to invest in [26]. Brogaard and Detzel (2015), on the other hand, mentioned that economic uncertainties affect stock prices [24]. Liu and Ren (2019) mentioned that some stocks are most sensitive to news than others [27]. Bleck and Liu (2007) revealed that the risk of a stock price crash stems from the continuous increase of the company's cost [28]. The impact of the COVID-19 pandemic has increased the uncertainty of global supply and demand. Many enterprises are stable in the futures market, leading to an unstable environmental business environment.

## 3. The research methods

### 3.1 Research data and methods

According to the abovementioned literature, this study intends to focus on the impact of the Wuhan City lockdown (2020/1/23-2021/4/7) on stock volatility during the outbreak of COVID-19 in China and analyze various scale indexes. The research data used include CSI All Share (Data source: China Securities Index Co., LTD.), CNI1000, CNI 2000, CNI Large Cap., CNI Mid-CAP., and CNI Small Cap. (Data source: A Wholly owned Subsidiary of Shenzhen Stock Exchange). the study duration was from 2017/7/1 to 2021, 1/6/30. Table 2 shows the six indicators that were used.

The trends of the six indexes are shown in Fig 1 (more constituent stocks) and Fig 2 (fewer constituent stocks).

Table 2. Research index introduction table.

| Index code | Index name | Number | E |
|---|---|---|---|
| 399303 | CNI 2000 | 2000 | Guoxin 2000 comprises 2000 A-shares with large market value and good liquidity. After deducting sample stocks of the Guoxin 1000 index from all A-shares, a price change trend in small and micro stocks in the A-share market was reflected. |
| 399311 | CNI 1000 | 1000 | CNI 1000 Index comprises the 1000 largest and most liquid A-share stocks listed and trading on the Shanghai and Shenzhen Stock Exchange. The index aims to reflect the performance of A-share large and middle-cap segments. |
| 399314 | CNI Large Cap. | 200 | CNI Large Cap. Index comprises the top 200 constituents from CNI 1000 Index as its candidate constituents. CNI Large Cap. Index provides a benchmark for the performance of large capital stocks. |
| 399315 | CNI Mid-Cap. | 300 | The Mega Tide Mid-Cap Index takes Guozi 1000 Index as the sample space to select the top 201–500 stocks with total market value, which reflects the operation of mid-cap stocks in the A-share market. |
| 399316 | CNI Small Cap. | 500 | Juchao Small Cap Index takes Guoshi 1000 Index as the sample space and selects the last 500 stocks with total market capitalization to reflect the operation of small-cap stocks in the A-share market. |
| 000985 | CSI All Share | indefinite | The CSI All Share Index includes all the A-shares except ST stocks, *ST stocks, and those that lasted less than three months. |

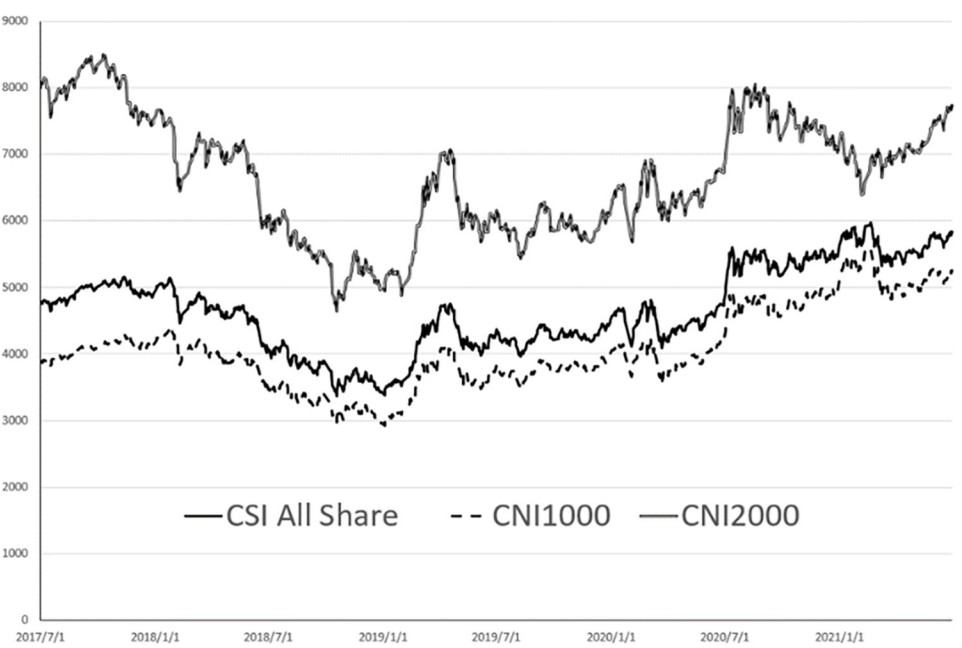

**Fig 1. CSI all share, CNI1000 and CNI2000 diagram.** CSI all share.

As shown in Fig 1, the trend amplitude of CNI 2000 (small stocks) is relatively large, ranging from 4637 to 8492, while the amplitude of CSI ALL Share and CNI 1000 is about 1000 points.

Fig 2 shows CNI Large Cap., CNI Mid-cap. And CNI Small Cap. The amplitude of the chart is between 3000 and 6000 points.

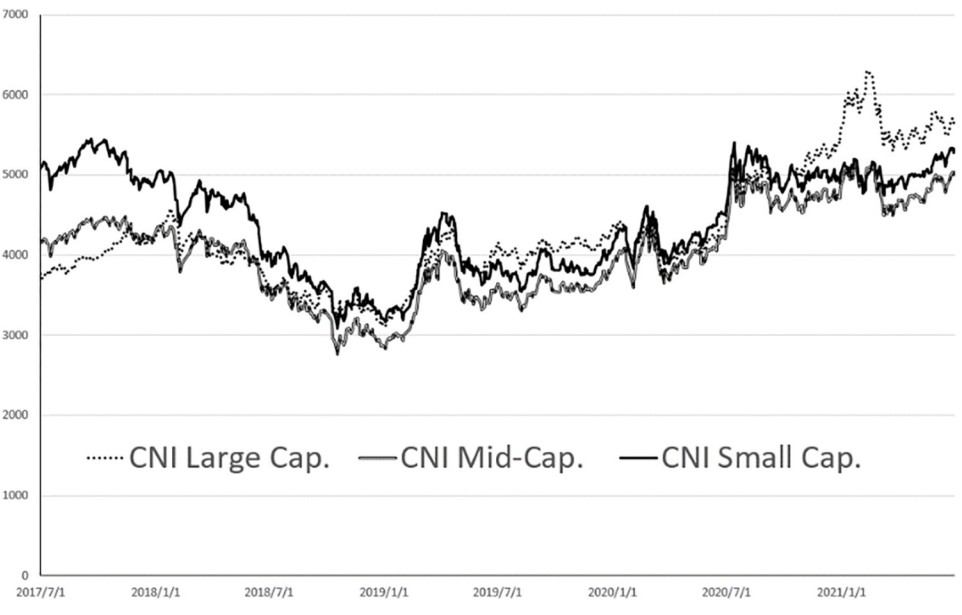

**Fig 2. CNI large cap, CNI mid-cap.** And CNI Small Cap Diagram.

In terms of holiday effects, this study used differences, i.e.

$$D = \begin{cases} 1 & \text{yes} \\ 0 & \text{no} \end{cases}$$

$$y = \beta_0 + \beta_1 D + \varepsilon$$

$$\frac{dy}{dD} = \beta_1$$

It is clear that $\beta 1$ is the difference between the two groups. This study used the difference-in-differences (DID) method, also known as the "double difference method", to investigate the impact of the COVID-19 pandemic on the stock holiday effect. As a policy, the economic effect evaluation method is a big weapon.

The model is as follows: $y_{it} = \beta_0 + \beta_1 d\mu + \beta_2 dt + \beta_3 d\mu \cdot dt + \varepsilon_{it}$

$$du = \begin{cases} 1 & \text{yes} \\ 0 & \text{no} \end{cases} \qquad dt = \begin{cases} 1 & \text{before} \\ 0 & \text{after} \end{cases}$$

$d\mu \cdot dt$ is the interaction term between the group dummy variable and the dummy variable before and after the event, $\frac{dy}{du} \cdot \frac{dy}{dt} = \beta_3$ reflecting the net effect of policy implementation.

For the robustness test of DID, the "placebo test" is generally used to divide the group before into two groups for further analysis. If the regression results of DID estimators are still significant under different fictions, it indicates that the original estimation results are likely biased.

For the use of relevant variables, the volatility (risk) of incoming stocks is generally replaced by the variables' standard deviation. Therefore, after the index is created, the variables under study must be transformed again. The model is as follows.

$$r_t = \ln\left(\frac{\text{Index}_t}{\text{Index}_{t-1}}\right) \times 100$$

$$r_t = \alpha_0 + \sum_{i=1}^{n} \alpha_i r_{t-i} + \varepsilon_t \qquad \varepsilon_t \sim^{iid} N(0, \sigma^2)$$

$$\text{Var}(r_t) = \text{Var}(\varepsilon_t)$$

$$h_t = \frac{\sqrt{\varepsilon_t^2}}{\mu_i}$$

$$h_t = \beta_0 + \beta_1 D + e_t$$

$$h_t = \delta_0 + \delta_1 D + \delta_2 T + \delta_3 T \cdot D + e_t$$

Obviously, $\beta$ is the holiday effect in this study, and $\delta 3$ is the net effect of the Wuhan lockdown and holiday effect.

## 3.2 The empirical analysis

This study examines six indexes, namely the market index (CSI All Share), the large-cap index (CNI1000), and the small-cap index (CNI2000). The CNI1000 was divided into CNI Large Cap., CNI Mid-cap., and CNI Small Cap. Using a time series analysis, the index itself is the CD root problem, so a reduction in the order is necessary. $r_t = \ln\left(\frac{\text{Index}_t}{\text{Index}_{t-1}}\right) \times 100$,

Table 3 shows the descriptive statistics of each index after the order reduction.

**Table 3. Descriptive statistics.**

|  | CSI ALL SHARE | CNI 1000 | CNI 2000 | CNI LARGE CAP. | CNI MID CAP. | CNI SMALL CAP. |
|---|---|---|---|---|---|---|
| Obs. | 973 | 973 | 973 | 973 | 973 | 973 |
| Mean | 0.02 | 0.03 | 0.00 | 0.04 | 0.02 | 0.00 |
| Median | 0.06 | 0.05 | 0.08 | 0.05 | 0.05 | 0.07 |
| Maximum | 5.44 | 5.69 | 4.77 | 5.76 | 5.68 | 5.44 |
| Minimum | -8.54 | -8.32 | -9.19 | -7.86 | -9.11 | -9.05 |
| Std. Dev. | 1.30 | 1.29 | 1.48 | 1.29 | 1.43 | 1.45 |
| Skewness | -0.69 | -0.58 | -0.93 | -0.41 | -0.77 | -0.80 |
| Kurtosis | 7.32 | 6.99 | 7.26 | 6.30 | 7.24 | 7.00 |

Table 3 illustrates that the absolute values of the maximum and minimum values of the six indices are larger than that of the minimum values, reflecting a larger decline amplitude. Moreover, all six indices present a left-skewed high isthmus peak pattern.

$$r_t = \alpha_0 + \sum_{i=1}^{n} \alpha_i r_{t-i} + \varepsilon_t$$

In the formula, the existence of a single root is most taboo, so the single root test must be performed. The results are shown in Table 4.

It can be determined from Table 4 that there is no single root in the six groups of data in this study, and the optimal formula can be judged $r_t = \alpha_0 + \sum_{i=1}^{n} \alpha_i r_{t-i} + \varepsilon_t$. Since the observed values in this study were all 973 (as shown in Table 2), Schwarz Criterion (SBC) is more appropriate for judging the optimal model. The SBC of each index lag stage is shown in Table 5.

It can be concluded from Table 5 that the most suitable model for the six indices is the model with a lag of one period, i.e.

$$r_t = \alpha_0 + \alpha_1 r_{t-1} + \varepsilon_t$$

Finally, it is generated according to the optimal model $h_t = \frac{\sqrt{\varepsilon_t^2}}{\mu_i}$.

This study focuses on the holiday effect, so it is designed to divide every other day off into four types and establish the dummy variables as follows Fig 3:

The net effect of the lockdown and holiday effect was calculated using two dummy variables, as shown below Fig 4.

So, two sets of models

$$h_i = \beta_0 + \beta_1 D_1 + \beta_2 D_2 + \beta_3 D_3 + e_i$$
$$h_i = \delta_0 + \delta_1 D + \delta_2 T + \delta_3 D \cdot T + \varepsilon_i$$

**Table 4. Single root test results.**

| Method | ADF | | PP | |
|---|---|---|---|---|
| Variable | t-Statistic | P value | t-Statistic | P value |
| CSI ALL SHARE | -30.91 | <0.001 | -30.91 | <0.001 |
| CNI 1000 | -31.09 | <0.001 | -31.10 | <0.001 |
| CNI 2000 | -29.71 | <0.001 | -29.68 | <0.001 |
| CNI LARGE CAP. | -30.93 | <0.001 | -30.93 | <0.001 |
| CNI MID CAP. | -31.04 | <0.001 | -31.04 | <0.001 |
| CNI SMALL CAP. | -30.68 | <0.001 | -30.68 | <0.001 |

**Table 5. Optimal schwarz criterion table (SBC).**

| Lag period according | CSI All Share | CNI 1000 | CNI 2000 | CNI Large Cap. | CNI Mid Cap. | CNI Small Cap. |
|---|---|---|---|---|---|---|
| AR(1) | 3.376 | 3.363 | 3.637 | 3.354 | 3.561 | 3.596 |
| AR(2) | 3.384 | 3.371 | 3.645 | 3.362 | 3.569 | 3.604 |
| AR(3) | 3.389 | 3.376 | 3.652 | 3.365 | 3.575 | 3.611 |
| AR(4) | 3.394 | 3.388 | 3.658 | 3.369 | 3.581 | 3.617 |
| AR(5) | 3.402 | 3.388 | 3.666 | 3.376 | 3.589 | 3.625 |
| AR(6) | 3.407 | 3.393 | 3.671 | 3.382 | 3.594 | 3.631 |

Table 6 shows that the constant coefficients of all kinds of indexes are positive (0.84, 0.84, 0.94, 0.85, 0.92, 0.94) and significant, indicating that stock volatility (risk) exists. $D_1$ (0.35, 0.32, 0.41, 0.27, 0.42, 0.42), $D_2$ (0.54, 0.52, 0.79, 0.43, 0.63, 0.71), and $D_3$(1.23, 1.24, 1.38, 1.24, 1.31, 1.32) coefficients are all positive and significant (except $D_2$ of CNI Large Cap.). The next day off is to increase the risk of stocks.

According to the data, the number of days off on the other day also seems to increase the risk of stocks. For the sake of rigor, this study establishes a statistical test table for illustration (as shown in Table 7).

Table 7 shows that the stock volatility (risk) increases as the number of days off increases. However, there is no significant difference between the three days off and the one or two days off. Meanwhile, the risk of more than four days off is significantly higher than that of less than two days off (significantly positive, 0.89, 0.92, 0.97, 0.97, 0.89, 0.90). The holiday effect of more than 4 days is significantly higher than the 3-day holidays, except for small stocks (CNI2000, CNI Small Cap), which are insignificant (0.59, 0.62).

Table 8 shows the double effect of the city lockdown, D represents the situation of the next day off, T is the time of the Windy City, and D*T is the net effect of city lockdown and vacation. The results show that the net effect of lockdown and holiday was significantly positive (1.65, 1.51, 1.98, 1.22, 1.93, and 1.87).

Placebo test can be used to determine whether the net effect of city lockdown and vacation is real. In this study, the placebo test was conducted during the same period. The results are shown in Table 9.

As shown in Table 9, D*T illustrates that most of the net effects of city lockdown and holidays are insignificant, indicating that the net effects of city lockdown and holidays exist. However, the large stocks (CNI1000, CNI Large Cap) are significant and negative, indicating that the net effects of city lockdown and holidays increase the risk of stocks. Therefore, the placebo test confirms that the overall effect of the lockdowns and holidays is to make stocks riskier.

$$(D_1, D_2, D_3) = \begin{cases} (0,0,0) & \text{隔天未放假} \\ (1,0,0) & \text{隔天放假一、二天} \\ (0,1,0) & \text{隔天放假三天} \\ (0,0,1) & \text{隔天放假四(含)天以上} \end{cases}$$

**Fig 3.**

| The original model | Placebo test |
|---|---|
| $du = \begin{cases} 1 & \text{隔天放假} \\ 0 & \text{其他} \end{cases}$ <br> $dt = \begin{cases} 1 & 2020/1/23 - 2020/4/7 \\ 0 & \text{其他} \end{cases}$ | $du = \begin{cases} 1 & \text{隔天放假} \\ 0 & \text{其他} \end{cases}$ <br> $dt = \begin{cases} 1 & 2018/1/23 - 2018/4/7 \\ 0 & \text{其他} \end{cases}$ |

**Fig 4.**

## 4. Conclusion

Many places in China have taken lockdown measures due to COVID-19. The lockdown may have curbed the spread of the virus, but it also created doubts among investors in the stock market. This study raises an interesting assumption on how lockdowns can create doubts among stock investors. This research focused on the design of environmental uncertainty, stock risk and emergency situation, social capital theory, and information theory to investigate the overall investment risk situation. Many investors have felt the psychological impact of the new coronavirus pandemic. This research study combined financial behaviors, marketing, investment, and other interdisciplinary disciplines to design and predict investors' behaviors and risk appetite during the COVID-19 pandemic.

One main contribution of this study is centered on the fact that no one has applied such a new research method. GARCH was only released in 1986. GARCH+DID is still completely unknown, making our research method more innovative.

Additionally, social capital theory and signal theory were used to determine the entire investment effect. The researchers combined psychology, financial behavior, and marketing across fields.

**Table 6. Vacation utility results table.**

| DV | CSI All Share | | CNI1000 | | CNI2000 | | CNI Large Cap. | | CNI Mid-Cap. | | CNI Small Cap. | |
|---|---|---|---|---|---|---|---|---|---|---|---|---|
| IV | Coefficient | | Coefficient | | Coefficient | | Coefficient | | Coefficient | | Coefficient | |
| | Std. Error | Significant condition | Std. Error | Significant condition | Std. Error | Significant condition | Std. Error | Significant condition | Std. Error | Significant condition | Std. Error | Significant condition |
| C | 0.84 | | 0.84 | | 0.94 | | 0.85 | | 0.92 | | 0.94 | |
| | (0.03) | *** | (0.03) | *** | (0.04) | *** | (0.03) | *** | (0.04) | *** | (0.04) | *** |
| D1 | 0.35 | | 0.32 | | 0.41 | | 0.27 | | 0.42 | | 0.42 | |
| | (0.07) | *** | (0.07) | *** | (0.08) | *** | (0.07) | *** | (0.08) | *** | (0.08) | *** |
| D2 | 0.54 | | 0.52 | | 0.79 | | 0.43 | | 0.63 | | 0.71 | |
| | (0.28) | * | (0.28) | * | (0.32) | ** | (0.28) | | (0.31) | ** | (0.31) | ** |
| D3 | 1.23 | | 1.24 | | 1.38 | | 1.24 | | 1.31 | | 1.32 | |
| | (0.23) | *** | (0.23) | *** | (0.27) | *** | (0.23) | *** | (0.25) | *** | (0.26) | *** |
| R2 | 0.05 | | 0.05 | | 0.05 | | 0.04 | | 0.05 | | 0.05 | |
| Adjusted R2 | 0.05 | | 0.04 | | 0.05 | | 0.04 | | 0.05 | | 0.05 | |
| F-statistic | 16.73 | | 16.09 | | 17.48 | | 14.48 | | 18.14 | | 18.12 | |
| Prob | <0.001 | | <0.001 | | <0.001 | | <0.001 | | <0.001 | | <0.001 | |

PS: *, ** and *** denote significance at the .1, .05 and .01 levels, respectively.

**Table 7. The next day holiday days different inspection table.**

| DV | CSI All Share | | CNI1000 | | CNI2000 | | CNI Large Cap. | | CNI Mid-Cap. | | CNI Small Cap. | |
|---|---|---|---|---|---|---|---|---|---|---|---|---|
| IV | Coefficient | | Coefficient | | Coefficient | | Coefficient | | Coefficient | | Coefficient | |
| | Std. Error | Significant condition | Std. Error | Significant condition | Std. Error | Significant condition | Std. Error | Significant condition | Std. Error | Significant condition | Std. Error | Significant condition |
| D1 | 0.35 | | 0.32 | | 0.41 | | 0.27 | | 0.42 | | 0.42 | |
| | (0.07) | *** | (0.07) | *** | (0.08) | *** | (0.07) | *** | (0.08) | *** | (0.08) | *** |
| D2 | 0.54 | | 0.52 | | 0.79 | | 0.43 | | 0.63 | | 0.71 | |
| | (0.28) | * | (0.28) | * | (0.32) | ** | (0.28) | | (0.31) | ** | (0.31) | *** |
| D3 | 1.23 | | 1.24 | | 1.38 | | 1.24 | | 1.31 | | 1.32 | |
| | (0.23) | *** | (0.23) | *** | (0.27) | *** | (0.23) | *** | (0.25) | *** | (0.26) | *** |
| D2-D1 | 0.20 | | 0.20 | | 0.37 | | 0.15 | | 0.20 | | 0.28 | |
| | (0.29) | | (0.29) | | (0.33) | | (0.29) | | (0.32) | | (0.32) | |
| D3-D1 | 0.89 | | 0.92 | | 0.97 | | 0.97 | | 0.89 | | 0.90 | |
| | (0.24) | *** | (0.24) | *** | (0.28) | *** | (0.24) | *** | (0.27) | *** | (0.27) | *** |
| D3-D2 | 0.69 | | 0.72 | | 0.59 | | 0.82 | | 0.68 | | 0.62 | |
| | (0.37) | * | (0.36) | ** | (0.42) | | (0.36) | *** | (0.40) | * | (0.41) | |

PS: *, ** and *** denote significance at the .1, .05 and .01 levels, respectively.

## 4.1. Limitation and future research

This study sorted out the data of small and large stocks in the Chinese stock market to observe their behaviors. In the empirical data analysis, researchers used Eviews to observe any changes. Future researchers are advised to use a syntax program to sort out data because Eviews require a lot of manpower and material resources for sorting data. Using programming syntax will make the research process much faster and guarantee accuracy. Moreover, future researchers should not focus on Chinese stock market analysis alone. To expand the research scope, future

**Table 8. Double difference table of lockdown and holiday effect.**

| DV | CSI All Share | | CNI1000 | | CNI2000 | | CNI Large Cap. | | CNI Mid-Cap. | | CNI Small Cap. | |
|---|---|---|---|---|---|---|---|---|---|---|---|---|
| IV | Coefficient | | Coefficient | | Coefficient | | Coefficient | | Coefficient | | Coefficient | |
| | Std. Error | Significant condition | Std. Error | Significant condition | Std. Error | Significant condition | Std. Error | Significant condition | Std. Error | Significant condition | Std. Error | Significant condition |
| C | 0.82 | | 0.82 | | 0.92 | | 0.83 | | 0.89 | | 0.91 | |
| | (0.03) | *** | (0.03) | *** | (0.04) | *** | (0.03) | *** | (0.03) | *** | (0.04) | *** |
| D | 0.34 | | 0.32 | | 0.40 | | 0.29 | | 0.40 | | 0.41 | |
| | (0.07) | *** | (0.07) | *** | (0.08) | *** | (0.07) | *** | (0.08) | *** | (0.08) | *** |
| T | 0.45 | | 0.44 | | 0.52 | | 0.45 | | 0.52 | | 0.54 | |
| | (0.15) | *** | (0.15) | *** | (0.17) | *** | (0.15) | *** | (0.16) | *** | (0.16) | *** |
| D*T | 1.65 | | 1.51 | | 1.98 | | 1.22 | | 1.93 | | 1.87 | |
| | (0.32) | *** | (0.32) | *** | (0.36) | *** | (0.31) | *** | (0.34) | *** | (0.35) | *** |
| R2 | 0.10 | | 0.09 | | 0.10 | | 0.07 | | 0.11 | | 0.11 | |
| Adjusted R2 | 0.09 | | 0.08 | | 0.10 | | 0.07 | | 0.11 | | 0.10 | |
| F-statistic | 34.68 | | 30.53 | | 37.44 | | 24.07 | | 40.01 | | 38.94 | |
| Prob | <0.001 | | <0.001 | | <0.001 | | <0.001 | | <0.001 | | <0.001 | |

PS: *, ** and *** denote significance at the .1, .05 and .01 levels, respectively.

**Table 9. Placebo test results table.**

| DV | CSI All Share | | CNI1000 | | CNI2000 | | CNI Large Cap. | | CNI Mid-Cap. | | CNI Small Cap. | |
|---|---|---|---|---|---|---|---|---|---|---|---|---|
| IV | Coefficient | | Coefficient | | Coefficient | | Coefficient | | Coefficient | | Coefficient | |
| | Std. Error | Significant condition | Std. Error | Significant condition | Std. Error | Significant condition | Std. Error | Significant condition | Std. Error | Significant condition | Std. Error | Significant condition |
| C | 0.83 | | 0.83 | | 0.94 | | 0.84 | | 0.91 | | 0.93 | |
| | (0.03) | *** | (0.03) | *** | (0.04) | *** | (0.03) | *** | (0.04) | *** | (0.04) | *** |
| D | 0.45 | | 0.43 | | 0.52 | | 0.38 | | 0.52 | | 0.52 | |
| | (0.07) | *** | (0.07) | *** | (0.08) | *** | (0.07) | *** | (0.08) | *** | (0.08) | *** |
| T | 0.14 | | 0.17 | | 0.12 | | 0.19 | | 0.21 | | 0.12 | |
| | (0.15) | | (0.15) | | (0.17) | | (0.15) | | (0.16) | | (0.17) | |
| D*T | -0.52 | | -0.59 | | -0.41 | | -0.56 | | -0.48 | | -0.34 | |
| | (0.33) | | (0.32) | * | (0.37) | | (0.32) | * | (0.36) | | (0.36) | |
| R2 | 0.04 | | 0.04 | | 0.04 | | 0.03 | | 0.04 | | 0.04 | |
| Adjusted R2 | 0.04 | | 0.03 | | 0.04 | | 0.03 | | 0.04 | | 0.04 | |
| F-statistic | 12.79 | | 11.98 | | 13.32 | | 9.78 | | 14.89 | | 14.29 | |
| Prob | <0.001 | | <0.001 | | <0.001 | | <0.001 | | <0.001 | | <0.001 | |

PS: *, **, and *** denote significance at the .1, .05 and .01 levels, respectively.

researchers can further analyze the stock market of different countries, such as Britain and the United States, Canada, and Australia, and investigate how the new coronary disease influenced many enterprises or industries in emergencies and investors' stock market investment during vacation.

## 5. Managerial and practical implications

This study is the earliest attempt to investigate the changes in the investment risks and stock market conditions of small and large stocks in China during the COVID-19 pandemic. This study focused on the impact of the COVID-19 pandemic on the stock effect and the overall difference between small and large stocks. The researchers developed new perspectives to address the future risk of investing in stocks by examining the combination of overall stock risk, environmental uncertainty, information theory, and social transaction theory. The researchers also developed a new model to contribute as follows. First, small and large stocks were affected after the holiday, affecting the stock market performance. Second, the pandemic affected the entire Chinese stock market, and environmental uncertainties led to volatile stock market performance. Third, investors made decisions about their subsequent investments based on the impact of the overall information they received.

## Author Contributions

**Conceptualization:** Li-Wei Lin.

**Formal analysis:** Shih-Yung Wei.

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
