## [Decision Letter · Decision Letter 0]

30 Jan 2023

PONE-D-22-30888The Impact of COVID-19 Environmental Uncertainty on The Perception of Investment StocksPLOS ONE

Dear Dr. Li-Wei Lin,

Thank you for submitting your manuscript to PLOS ONE. After careful consideration, we feel that it has merit but does not fully meet PLOS ONE’s publication criteria as it currently stands. Therefore, we invite you to submit a revised version of the manuscript that addresses the points raised during the review process.

We look forward to receiving your revised manuscript.

Kind regards,

Ricky Chee Jiun Chia

Academic Editor

PLOS ONE

Journal Requirements:

"No"

"We have no conflict of interest."

Reviewers' comments:

Reviewer's Responses to Questions

**Comments to the Author**

1. Is the manuscript technically sound, and do the data support the conclusions?

Reviewer #1: Partly

Reviewer #2: Yes

2. Has the statistical analysis been performed appropriately and rigorously? 

Reviewer #1: Yes

Reviewer #2: Yes

3. Have the authors made all data underlying the findings in their manuscript fully available?

Reviewer #1: Yes

Reviewer #2: Yes

4. Is the manuscript presented in an intelligible fashion and written in standard English?

Reviewer #1: No

Reviewer #2: Yes

5. Review Comments to the Author

Reviewer #1: This paper examines a version of the day of the week/weekend effect anomaly studies that were a common feature of the finance literature in the 1980's. Recent authors have suggested that the anomaly has largely dissipated in North America, possibly because traders take it into consideration. While the current study's results are consistent with the authors' expectations, the work is pedestrian and very basic. I'm not convinced that it would be of much interest to researchers despite the topicality of Covid-19. My main concern with the paper though is that it is barely intelligible and requires a very thorough rewriting by someone who knows the subject and whose first language is English. Beyond the exposition, the paper is constructed poorly, which, even if one struggles through the exposition, makes it difficult to understand where the authors are going. In addition, the title is unhelpful inasmuch as it gives no indication that the paper is about the "holiday effect'" in China. The motivation (why this work is important) is not well explained and inadequate attention is given to explaining why multiple indices are examined and why the results among indices differ. Finally, the construction of the Bibliography is sloppy, with incomplete references and inconsistencies in formatting.

Reviewer #2: This is a very interesting paper and my only concern for the paper is that the author(s) should give us more solid arguments about their empirical findings. What are their contributions compared to previous studies? All these need to be clarified in the revised version of paper.

6. PLOS authors have the option to publish the peer review history of their article (what does this mean?). If published, this will include your full peer review and any attached files.

Reviewer #1: No

Reviewer #2: **Yes: **TSANG YAO CHANG

---

## [Author Response · Author response to Decision Letter 0]

31 Mar 2023

Thanks for your suggestion.

1. We have strengthened the literature sources for 2016-2023 in the literature.

2. Our research design is rigorous, including strict design in content, structure, and method, and has certain quality standards in publishing SSCI.

3. In the article, we use the red part to indicate the modified part

4.We let professional English teacher Jake to participate and help.

www.academiccommunications.com.

The English is in line with the sentences and English standards published in SSCI.

---

## [Editor Report · Decision Letter 1]

6 Apr 2023

Impacts of Environmental Uncertainty on Investment Stocks Perception under the Holiday Effect

PONE-D-22-30888R1

Dear Dr. Li-Wei Lin,

We’re pleased to inform you that your manuscript has been judged scientifically suitable for publication and will be formally accepted for publication once it meets all outstanding technical requirements.

Kind regards,

Ricky Chee Jiun Chia

Academic Editor

PLOS ONE
---

## [Editor Report · Acceptance letter]

12 Apr 2023

PONE-D-22-30888R1 

Impacts of Environmental Uncertainty on Investment Stocks Perception under the Holiday Effect 

Dear Dr. Lin:

I'm pleased to inform you that your manuscript has been deemed suitable for publication in PLOS ONE. Congratulations! Your manuscript is now with our production department. 

Kind regards, 

on behalf of

Dr. Ricky Chee Jiun Chia 

Academic Editor

PLOS ONE